# Process Optimization of Robotic Polishing for Mold Steel Based on Response Surface Method

Yinhui Xie [1], Guangsheng Chang [1,2], Jinxing Yang [1], Mingyang Zhao [1] and Jun Li [1,*]

1 Quanzhou Institute of Equipment Manufacturing Haixi Institutes, Chinese Academy of Sciences, Quanzhou 362200, China; xyh1932@fjirsm.ac.cn (Y.X.); changuansen@163.com (G.C.); jxyang@fjirsm.ac.cn (J.Y.); 218527144@fzu.edu.com (M.Z.)
2 Beijing Key Laboratory of Advanced Manufacturing Technology, Beijing University of Technology, Beijing 100124, China
* Correspondence: junli@fjirsm.ac.cn

**Abstract:** Aimed to reduce surface roughness (Ra) and improve surface quality of mold steel, the optimizations of process parameters for robotic polishing, such as polishing pressure, feed speed and rotating speed of tool, are accomplish in this research. The optimum range of each parameter is obtained according to a single factor experiment, and the central composite design experiments on the three polishing parameters are conducted to establish a prediction model of surface roughness. Furthermore, a significance test of the prediction model is carried out through variance analysis. The optimum polishing parameters are obtained based on the analysis of response surface, and are then adopted in the polishing experiments of mold steel for validation. The experiment result of model verification indicates that the relative errors of predicted Ra ratio and actual Ra ratio are within the allowable range (maximum is 13.47%). It proves the accuracy of the roughness prediction model. Meanwhile, the experimental results of multipath polishing show that the surface roughness decreased effectively after polishing with the optimum polishing parameters. The prediction model of surface roughness and optimum polishing parameters are helpful to improve surface quality in robotic polishing for mold steel.

**Keywords:** robotic polishing; mould steel; surface roughness; response surface method; parameters optimization

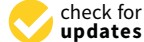



## 1. Introduction

The surface quality of a mold has a direct impact on the quality and service life of mold casting products. In order to remove the knife marks and hard layers left by mechanical machining on the surface of a mold cavity, the die surface needs to be polished to obtain the required size, shape and surface roughness. Mold steel is an important technical and material base of die industry and is used in the manufacture of various dies. Additionally, its surface quality has great influence on the performance, service life and manufacturing cycle of die [1,2].

At present, most polishing process of mold steel is carried out by hand, which takes more than 20% of the total mold manufacturing time and 30% of the total product machining cost. In addition, there are many defects such as poor polishing accuracy, low efficiency, lack of consistency and stability of product quality in manual polishing and the process must be carried out by skilled workers, resulting in high production costs and long work cycle [3]. A lot of achievements have been made on the automatic polishing technology of mold steel, both at home and abroad. Shan et al. [4] analyzed the thermodynamic process in laser polishing based on the basic equation of heat conduction. The quasi-static model is established by numerical simulation to predict the mold steel process parameters of laser polishing. Wang et al. [5] used magnetorheological polishing fluid to polish the aspheric surface made of S-136 mold steel. The effects of polishing parameters, including

constituent of magnetorheological polishing fluid and rotational speed of workpiece on surface roughness Ra and surface forming, were researched. Almeidaa et al. [6] predicted the material removal in the course of polishing mold steel. The simulation model eliminated the repetitive machining and polishing error, therefore promoting the production of mold steel.

The machining efficiency and surface quality of mold steel should be considered at the same time. Based on the above presented state of automatic polishing technology for mold steel, it should be noted that optimization of the polishing process parameters, especially, are reflected in the study of surface roughness. However, there are still little research on the polishing process parameters (especially polishing pressure) and surface roughness prediction of the robotic polishing for mold steel. Ma et al. [7] built a prediction model of surface roughness for point grinding low expansion glass by applying BP neural networks and genetic algorithm, and optimization of parameters was carried out based on genetic algorithm. Junde et al. [8] investigated the coupling influence of process parameters such as sand belt speed, maximum cutting depth and feed speed on surface roughness through an artificial neural network method, and built a prediction model of surface roughness. For grinding 45# steel with a sand belt, Li et al. [9] established a prediction model of surface roughness with the response surface method and obtained the optimal value of process parameters. Perec et al. [10] adopted the methodology of the response surface to create a mathematical-statistic model about erosion of the metamorphic rock—marble by the abrasive water jet. Perec et al. [11] built the model of abrasive water suspension jet cutting process through using a response surface method. The best dimensions of the working nozzle and level of abrasive flow rate were obtained to achieve the biggest cutting depth. Nguyen et al. [12] optimized the machining parameter of shear thickening polishing for gear surfaces by response surface methodology. The method could improve the surface quality and mechanical properties of material. Das et al. [13] conducted research into process parameters and characterization of surface texture with rotational-magnetorheological abrasive flow finishing through using response surface methodology. The experimental result showed that rotational speed of the magnet had a significant effect on output response. The research results of this research shows that the response surface method can be applied to process parameter optimization in different processing scenarios.

In the above prediction model of surface roughness, a variety of prediction methods were used. The genetic algorithm takes all individuals in a population as an object and guides the efficient search of a coded parameter space by using the randomization. However, the genetic algorithm has certain dependence on the selection of initial population and low search efficiency. The neural network system is a complicated dynamic system with highly non-lineal kinetics. The neural network needs a lot of parameters and is unable to observe the learning process, that resulted in studying for too long may not even achieve the purpose of learning [14]. The response surface method establishes a continuous variable surface model to evaluate the factors and their interactions affecting processes and determine the optimal level range. Moreover, the number of experimental groups is relatively small, which can improve the efficiency of the optimization calculation. Compared with the genetic algorithm and the neural network algorithm, the response surface method has higher efficiency and fitting precision [15].

The polishing pressure varies greatly due to the vibration and the change of workpiece surface curvature. The control of polishing pressure is always the focus of polishing process optimization. Aimed to accurately explore the influence of polishing pressure, rotating speed of the tool and feed speed on the quality of the workpiece's surface during the polishing process, a robotic polishing platform with constant force control is built based on a 6-DOF industrial robot and a six-dimensional force sensor. In this research, the response surface method is adopted to optimize the process parameters of robotic polishing. Then the polishing parameters experiments of mold steel are tested with a small polishing tool. The regression analysis of polishing parameters is carried out by the response surface method,

and the prediction model of surface roughness is established to realize the prediction and control of surface roughness.

## 2. Experiments Preparation

### 2.1. Experiment Equipment Setup

For the purpose of studying the influence of polishing pressure variation on machining quality, it is necessary to control the size and direction of polishing pressure during the polishing process. Therefore, a robotic polishing platform with force control is established, as in Figure 1. The robotic polishing platform mainly consists of three main parts: industrial robot, magnetic worktable for polishing and circulating water cooling system [16–18]. The model of industrial robot is KUKA KR60-3 with a 60 kg load. Additionally, the repositioning accuracy of the robot is ±0.06 mm. The positioning of the polishing tool is carried out by the movement of a manipulator. The magnetic worktable is installed in the working range of an industrial robot and used for fixing the workpiece. The circulating water-cooling system can provide cooling polishing fluid through a double circulation channel during workpiece processing. Figure 2 gives the structure of the end-effector, which is composed of a small polishing tool, servo motor and six-dimensional force sensor, etc. The servo motor rotates the polishing tool to realize the polishing of the workpiece surface, and the six-dimensional force sensor is adopted to collect the variations of polishing force during the course of polishing. The six-dimensional force sensor is installed at the end of manipulator of the model, which is ATI Delta IP60. In the coordinate system of the force sensor, the range of pressure in the $X$ and $Y$ directions is ±330 N and the range of pressure in the $Z$ direction is ±990 N. The range of torque in the $X$, $Y$ and $Z$ directions are all 30 N·m. Additionally, a high-speed servo motor is selected as the spindle motor to provide the rotation speed of the polishing tool. The servo motor which is the DELTA ECMA-C10604R model has a power of 400 W and a Max RPM of 4500. During the course of polishing, the forces between the workpiece and the polishing tool are collected for transmission to the controller of the robot. By combining the constant control of poshing pressure, the polishing tool is precisely positioned through the 6-DOF industrial robot.

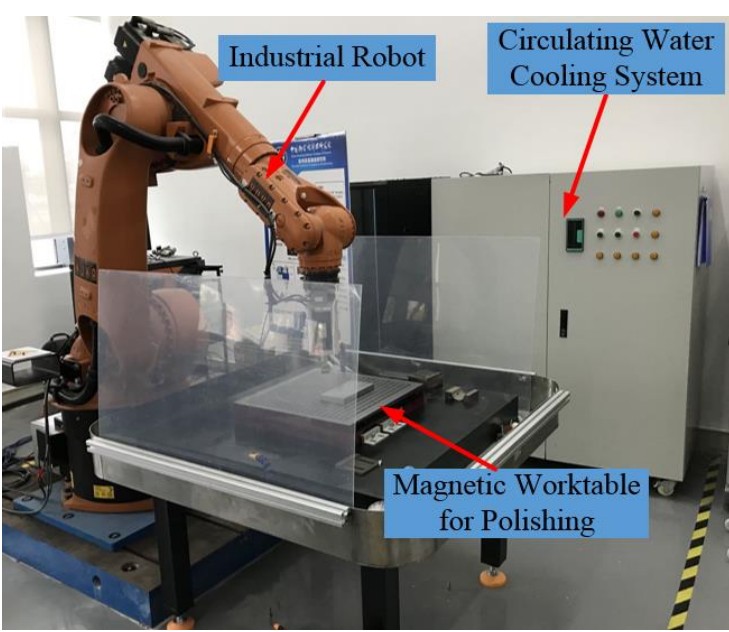

**Figure 1.** The robotic polishing platform with force control.

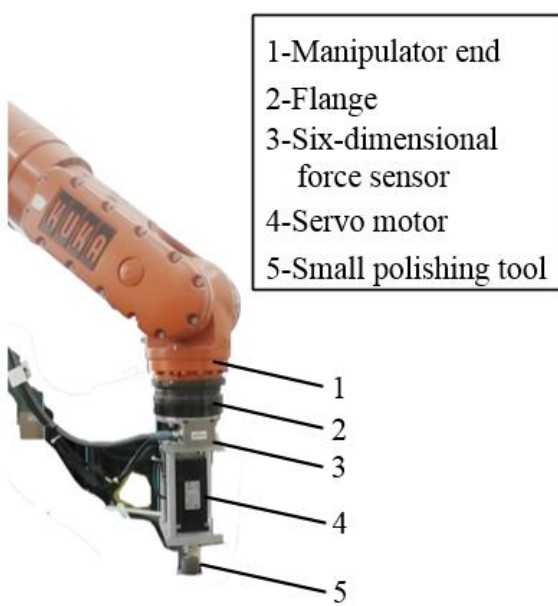

**Figure 2.** The end-effector of the robot.

*2.2. Experiment Conditions*

The cylindrical tool and the plane mold steel are selected as the polishing tool and workpiece, respectively, as shown in Figure 3. The polishing tool, which has a radius of 16.5 mm is made of nylon, and the material of mold steel is 40Cr, which has a specific chemical composition that includes: carbon C: 0.45%, Silicon Si: 0.37%; Manganese Mn: 0.8%; Chromium Cr: 1.1%; Iron Fe: >90%. Table 1 shows the specific material parameters. The polishing fluid is an aqueous solution of 2000 mesh white corundum which has a concentration of 5%, and the abrasive is a diamond polishing paste, which has a brand of W10 (2000 mesh). The surface roughness Ra of mold steel is measured by a stylus surface roughness measuring instrument (SJ-210). The diameter of tip is 2 μm, as shown in Figure 4. The speed of the measurement is 0.5 mm/s and the time of each measurement is 15 s.

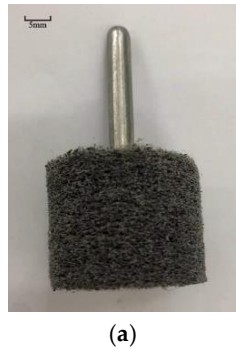

(**a**)

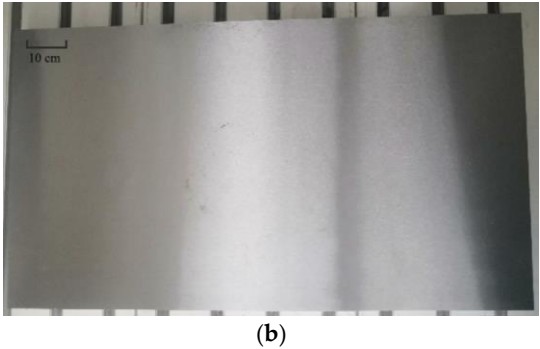

(**b**)

**Figure 3.** The polishing tool and the initial mold steel: (**a**) Polishing tool; (**b**) Plane mold steel.

**Table 1.** The material parameters.

| Name | Material | Elastic Modulus (GPa) | Poisson Ratio | Density (kg/m$^3$) |
|---|---|---|---|---|
| Mold steel | 40Cr | 211 | 2.77 | 7850 |
| Polishing tool | Nylon | 8.3 | 0.28 | 1150 |

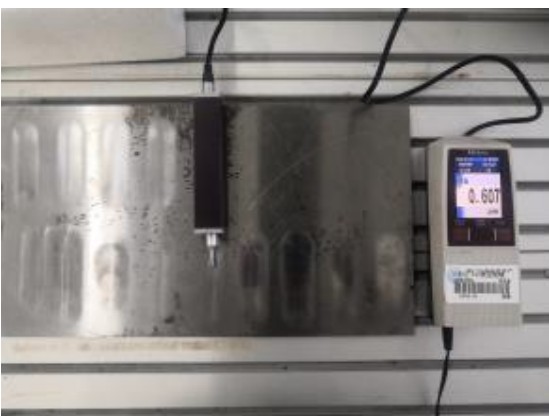

**Figure 4.** The measurement process of roughness.

## 3. Parameters Optimization of Single Polishing Path

Usually, the raster path is chosen to be the machining path for the robotic polishing process of the plane. In order to explore the effect of several process parameters on surface roughness of mold steel quickly, the first experiments in a single path are performed to optimize the polishing parameters of a single path. Then the multipath machining experiments are carried out by determined the distance of the raster path. The key parameters of the single-path polishing experiment are selected for the single-factor experiments, and the center composite experiments are designed according to the result of single-factor experiments. It can not only reduce the number of experimental groups but also obtain more accurate combination of optimal process parameters.

### 3.1. Determination of Process Parameters Level

The process parameters that have a great influence on surface roughness, including polishing pressure, rotation speed of tool and feed speed were selected as research objects. With these parameters as constraint conditions and the surface roughness as the optimization objective, the single factor experiments were carried out. Based on the actual condition of polishing, the values of polishing parameters are selected, as shown in Table 2. Where $P$, $R$ and $V$ represent polishing pressure, rotation speed of tool and feed speed, respectively. The initial average roughness of experimental samples is 1.384 μm. When the single factor experiments are carried out, the other two factors are set as the middle value of the variable range, respectively apart from the variable factor. Additionally, the three different points of surface were measured three times in the middle of the polishing path at the same processing condition to calculate the average value which was taken as the surface roughness after polishing. According to the experimental results, the relationships between polishing pressure, rotation speed of tool, feed speed and surface roughness of mold steel can be obtained, respectively, as shown in Figure 5.

Based on the result of single factor experiments, the optimum range of polishing parameters is determined. The ranges of polishing pressure, rotation speed of tool and feed speed are 10~50 N, 500~2500 r/min and 0.25~1.25 mm/s, respectively. Moreover, the experimental results show that the changes of key process parameters have obvious influence on the surface roughness.

**Table 2.** Polishing parameters of single factor experiment.

| Parameter | Value | | | | |
|---|---|---|---|---|---|
| $P$ (N) | 10 | 20 | 30 | 40 | 50 |
| $R$ (r/min) | 500 | 1000 | 1500 | 2000 | 2500 |
| $V$ (mm/s) | 0.25 | 0.50 | 0.75 | 1.00 | 1.25 |

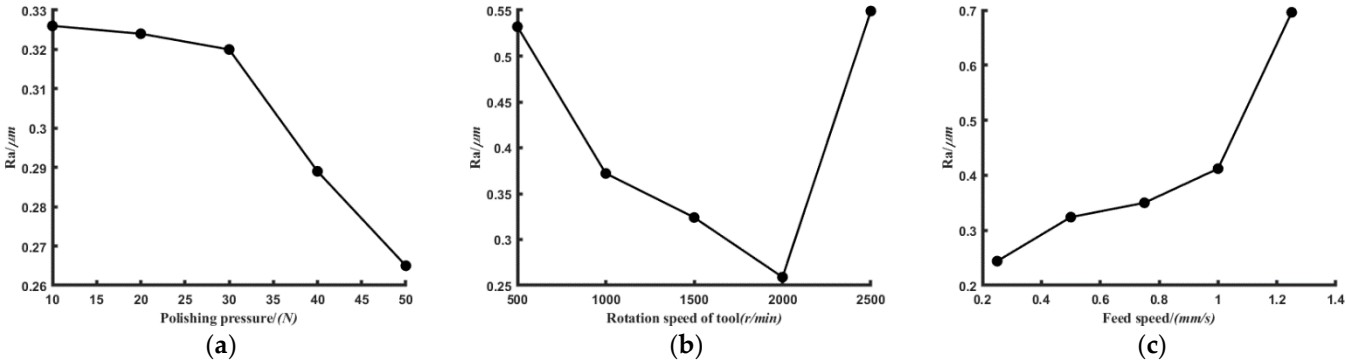

**Figure 5.** The relationships between process parameters and surface roughness: (**a**) Polishing pressure; (**b**) Rotation speed of tool; (**c**) Feed speed.

### 3.2. Center Composite Design Experiment

In the optimization of polishing process parameters, the central composite design method is an experimental design method with high recognition and application. It can evaluate the linear and interactive terms as well as the high-order surface effect and provide more effective data of the independent variables and errors with the least number of test cycles [19]. Three levels of each parameter are chosen and the distribution of the polishing parameter is built according to the optimum range of polishing parameters, as shown in Table 3. The central composite design experiment was planned for 17 groups through Design Expert software. Table 4 shows the parameters setting of the experiment and measurement results. Additionally, the surface of mold steel after polishing is shown in Figure 6.

**Table 3.** Levels and distribution of polishing parameters.

|  |  | Level | | |
|---|---|---|---|---|
| **Parameter** |  | **−1** | **0** | **+1** |
| **A** | $P$ (N) | 10 | 30 | 50 |
| **B** | $R$ (r/min) | 500 | 1500 | 2500 |
| **C** | $V$ (mm/s) | 0.25 | 0.75 | 1.25 |

**Table 4.** Experiments and results using central composite design. Ra-surface roughness after polishing; $Ra_0$-Initial surface roughness; $y$-Ratio of Ra and $Ra_0$.

| No. | A-$P$ (N) | B-$R$ (r/min) | C-$V$ (mm/s) | Ra (μm) | $Ra_0$ (μm) | $y$ |
|---|---|---|---|---|---|---|
| **1** | 30 | 500 | 0.75 | 0.814 | 1.606 | 0.507 |
| **2** | 50 | 1500 | 0.25 | 0.328 | 1.748 | 0.187 |
| **3** | 50 | 1500 | 1.25 | 0.751 | 1.762 | 0.426 |
| **4** | 10 | 500 | 1.25 | 1.384 | 1.439 | 0.962 |
| **5** | 30 | 1500 | 0.25 | 0.497 | 1.946 | 0.155 |
| **6** | 50 | 2500 | 0.75 | 0.698 | 1.872 | 0.373 |
| **7** | 50 | 1500 | 1.25 | 0.759 | 1.663 | 0.457 |
| **8** | 10 | 1500 | 0.75 | 0.816 | 1.469 | 0.555 |
| **9** | 30 | 1500 | 0.25 | 0.171 | 1.517 | 0.253 |
| **10** | 50 | 500 | 0.25 | 0.418 | 1.677 | 0.249 |
| **11** | 10 | 2500 | 0.25 | 0.433 | 1.579 | 0.274 |
| **12** | 10 | 1500 | 0.75 | 0.815 | 1.721 | 0.474 |
| **13** | 30 | 2500 | 1.25 | 1.135 | 1.865 | 0.609 |
| **14** | 30 | 2500 | 1.25 | 1.167 | 1.881 | 0.62 |
| **15** | 30 | 500 | 0.75 | 1.031 | 1.909 | 0.54 |
| **16** | 10 | 500 | 0.25 | 0.983 | 1.725 | 0.57 |
| **17** | 10 | 1500 | 0.25 | 0.33 | 1.956 | 0.169 |

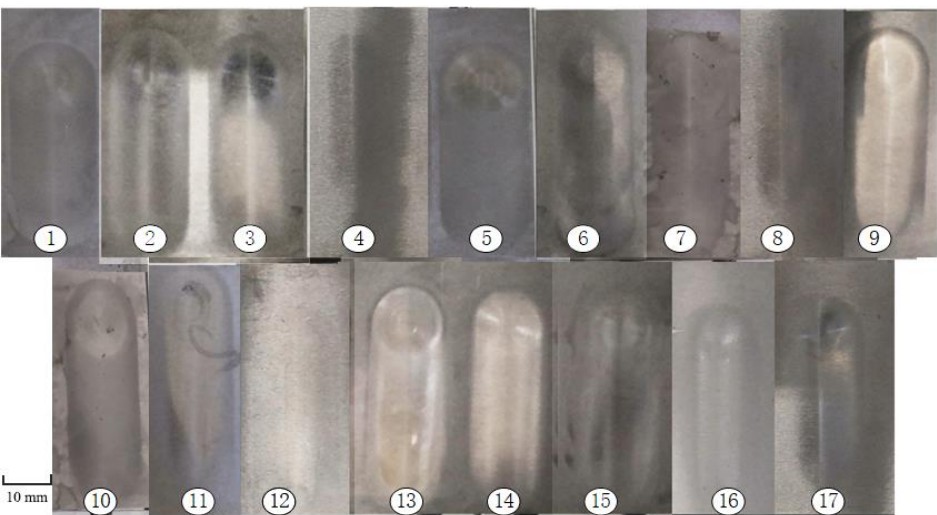

**Figure 6.** The surface of mold steel after polishing.

Due to the different settings of polishing parameters, the differences of polishing effects can be seen from Figure 6. When the polishing pressure is greater than 30 N, the polishing traces of mold steel are more obvious. In addition, the surface quality of mold steel is also related to rotation speed of tool and feed speed.

In the workpiece polishing process, surface roughness of workpiece is affected by many polishing parameters. The inputs and outputs in the prediction model of surface roughness are not proportional. Therefore, the prediction model of surface roughness is nonlinear and can be derived as Equation (1) by using quadratic regression [20].

$$Y = R_a - \varepsilon = b_0 + \sum_{i=1}^{k} b_i x_i + \sum_{i=1}^{k} \sum_{j=1, i<j}^{k} b_{ij} x_i x_j + \sum_{i=1}^{k} b_{ii} x_i^2 \tag{1}$$

where $Y$ is the surface roughness of prediction model, $R_a$ is the surface roughness after polishing, $\varepsilon$ is the experimental error, $b$ is the coefficients of model and $x$ is the polishing parameter in each level.

The coefficient matrix of the prediction model can be obtained through the following equation:

$$b = (X^T X)^{-1} X^T Y_1 \tag{2}$$

where $X$ is the matrix comprised of the experimental variables and $Y_1$ is the matrix composed by the experimental results.

### 3.3. Significance Test of Prediction Model

Aiming to improve the reliability of the prediction model, it is necessary to eliminate the insignificant factors in the model. Expert Design software was used to conduct variance analysis on the experimental data in Table 4. Additionally, the insignificant factors in the model were eliminated according to the step-by-step selection method, thus obtaining the prediction model. Table 5 shows the analysis results.

Variance analysis is mainly to test the fitting accuracy of the model, and the commonly used indicators include model testing value $F$, model probability value $p$ and coefficient of determination $R^2$. The larger the testing value $F$ is and the smaller the probability value $p$ is in the model, the fitting accuracy of the model is higher. The model testing value $F$ of 19.54, which is greater than the critical value $F_c$ of 3.74, illustrates the high significance of the model.

The model testing value $F$ with a model probability value $p$ (0.0004), which is near to zero explains the high significance of this regression model. The coefficient of determination $R^2$ is calculated to be 0.9570 for response. The calculation result of $R^2$ (coefficient of

determination) indicates that 95.7% of experiment the data are in agreement with the data of the predicted model, and only 4.3% of the total variations are found in the model. The value of Adjusted $R^2$ (adjusted coefficient of determination) is 0.9312 and shows that the model has high significance. The Predicted $R^2$ is 0.8502 and indicates that 85% of the variability in the prediction model could be explained. Additionally, the Predicted $R^2$ is consistent with the Adjusted $R^2$ of 0.9312. Adeq precision represents the signal to noise ratio. Additionally, the model ratio of 22.9607, which is greater than 4 indicates sufficient signal. So, the prediction model can be considered reliable.

**Table 5.** Analysis of experimental variance. $R^2$ (coefficient of determination) = 0.9570, Adjusted $R^2$ = 0.9312, Predicted $R^2$ = 0.8502, Adeq precision = 22.9607, *DOF* degrees of freedom, *F* testing value, *p* probability value, [S] Significant, [N] Not significant.

| Variation Source | Quadratic Sum | *DOF* | Root Mean Square | *F* | *p* |
|---|---|---|---|---|---|
| Model | 0.6744 | 9 | 0.0749 | 19.54 | 0.0004 [S] |
| A-*P* [S] | 0.0981 | 1 | 0.0981 | 25.59 | 0.0015 |
| B-*R* [S] | 0.0283 | 1 | 0.0283 | 7.38 | 0.0299 |
| C-*V* [S] | 0.3209 | 1 | 0.3209 | 83.68 | <0.0001 |
| AB [S] | 0.0200 | 1 | 0.0200 | 5.22 | 0.0563 |
| AC [S] | 0.0160 | 1 | 0.0160 | 4.17 | 0.0804 |
| BC [N] | 0.0004 | 1 | 0.0004 | 0.1031 | 0.7575 |
| A $^{2}$[S] | 0.0022 | 1 | 0.0022 | 0.5691 | 0.4752 |
| B $^{2}$[S] | 0.0416 | 1 | 0.0416 | 10.86 | 0.0132 |
| C $^{2}$[N] | 0.0008 | 1 | 0.0008 | 0.1963 | 0.6711 |
| Residual error | 0.0268 | 7 | 0.0038 | | |
| Lack of fit | 0.0177 | 2 | 0.0088 | 4.82 | 0.0682 [N] |
| Pure error | 0.0092 | 5 | 0.0018 | | |
| Cor total | 0.7013 | 16 | | | |

As can be seen in Table 5, the model testing value *F* of the single-factor polishing pressure A, rotation speed of tool B, feed speed C and the interaction terms AB, AC and B$^2$ in the prediction models are all greater than 3.74, which are significant to the response value, so they should be retained. However, the test values *F* of interaction terms BC, A$^2$ and C$^2$ are less than 3.74, which has no significant influence on the response value, so it should be removed. Then, the actual prediction model is as follows after eliminating the insignificant factors:

$$R_a = 0.658 - 7.05 \times 10^{-3}A - 4.83 \times 10^{-4}B + 0.503 \times 10C \\ + 3.558 \times 10^{-6}AB - 4.62 \times 10^{-3}AC + 1.05 \times 10^{-7}B^2 \tag{3}$$

The diagnosis test of the prediction model is conducted through Design-Expert fitting, and the result of residual analysis is presented in Figure 7. Most of the scattered points of residuals fall around the predicted value and deviate from the straight line to a small degree. Therefore, it can be considered that residuals follow normal distribution and the model fits well. Figure 8 shows that the predicted data are mainly in agreement with experimental result and the prediction model can achieve a high precision prediction effect by comparison of predicted data with experimental result.

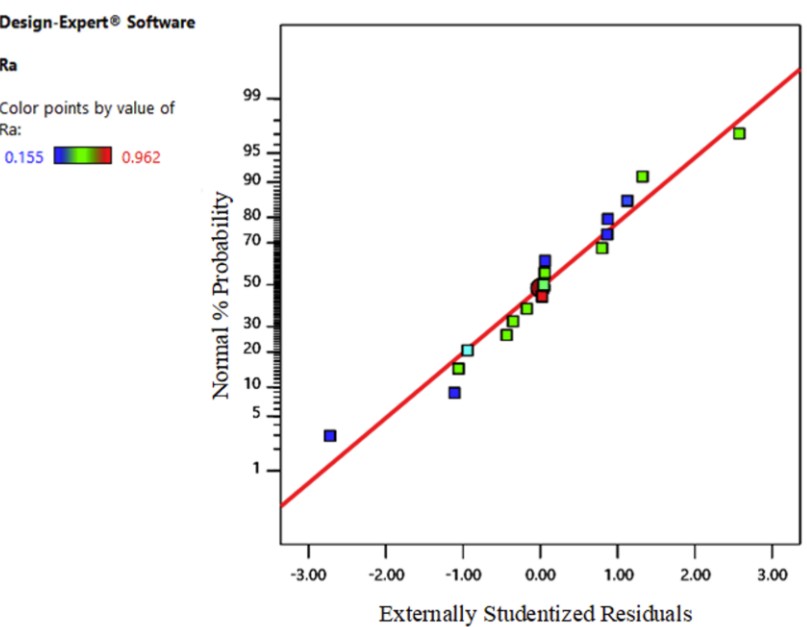

**Figure 7.** Normal plot of residuals.

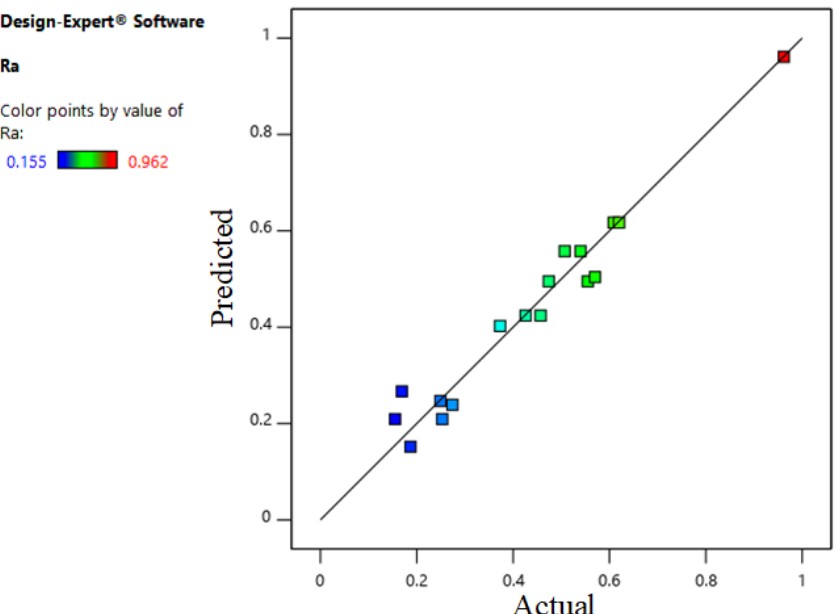

**Figure 8.** The comparison of predicted data with experimental result.

*3.4. Optimization and Validation of Process Parameters*

According to the results in Table 4, the response surfaces of polishing process parameters are fitted as shown in Figure 9. Figure 9a is the response values of feed speed and polishing pressure for surface roughness. The response surface curvature is small due to the significant difference of interaction for feed speed and polishing pressure during the process of polishing. The surface roughness decreases with the increase in feed speed, and the roughness value decreases more when the polishing pressure is 50 N. While the roughness value decreases less when the polishing pressure is 10 N. The roughness value decreases greatly when the feed speed is larger. Thus, it can be judged that when the feed speed is 0.25 mm/s the surface roughness value decreases more. Figure 9b is the response values of rotation speed of the tool and polishing pressure for surface roughness. According to the response surface, when the polishing pressure is 50 N, the surface roughness decreases

firstly and then increases with the increase in the rotation speed. When the rotation speed of the tool is about 1600 r/min, the surface roughness is the lowest. Therefore, the optimal process parameters for polishing are a polishing pressure of 50 N, rotation speed of the tool of 1600 r/min and feed speed of 0.25 mm/s.

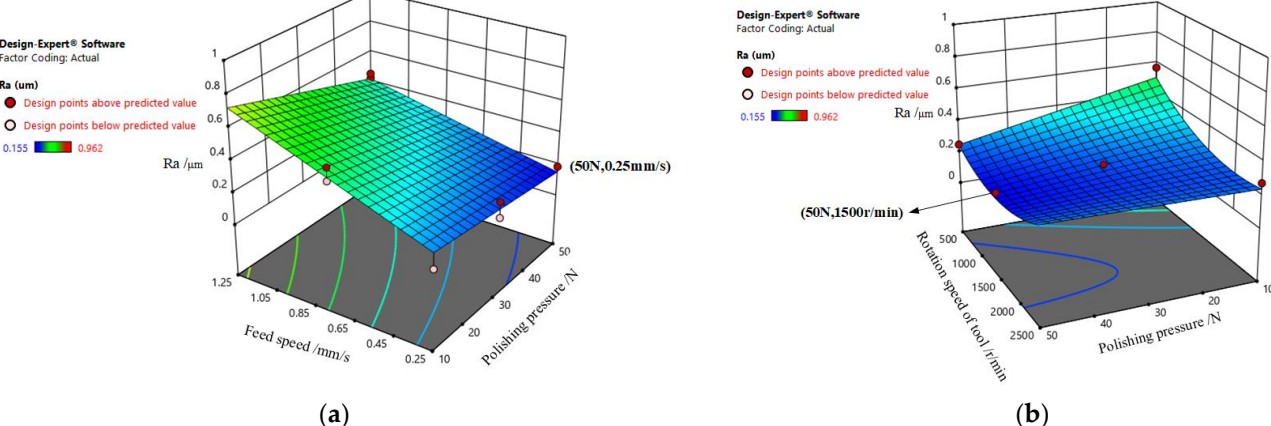

**Figure 9.** The response surfaces of polishing process parameters. (**a**) The response values of feed speed and polishing pressure; (**b**) The response values of rotation speed of tool and polishing pressure.

The process parameters are optimized according to the response surfaces in process parameter selection range of prediction model. The minimum Ra (surface roughness) ratio and the maximum desirability are set as the optimization goal, and 5 groups of optimal parameters combinations are obtained by using Design Expert software optimization, as shown in Table 6. The roughness ratio is the value of roughness after machining to initial roughness. The Ra ratio is the value of Ra after polishing to initial Ra and the desirability of which is a maximum of 1 which represents the reliability of the parameter combination in the response surface. The more the desirability approaches 1, the higher the reliability. The optimal parameters combinations can be used in whole surface polishing of mold steel.

**Table 6.** The optimal parameters combinations.

| No. | Polishing Pressure (N) | Rotation Speed of Tool (r/min) | Feed Speed (mm/s) | Ra Ratio | Desirability |
|-----|-----|-----|-----|-----|-----|
| **1** | 50 | 1624 | 0.25 | 0.254 | 0.946 |
| **2** | 50 | 1644 | 0.25 | 0.215 | 0.944 |
| **3** | 50 | 1622 | 0.26 | 0.256 | 0.941 |
| **4** | 50 | 1417 | 0.25 | 0.274 | 0.941 |
| **5** | 50 | 1403 | 0.25 | 0.233 | 0.940 |

For the purpose of verifying the accuracy of the prediction model of the surface roughness, five groups of process parameters combinations were randomly selected for verification of the polishing experiments (as shown in Table 7). The polishing experiment results of mold steel surface are shown in Figure 10. Additionally, the measurement result of Ra ratio is shown in Table 7. The surface quality polished by using the optimized parameters (No.5) is obviously better than that polished by a random combination of process parameters. Figure 11 shows the comparison between predicted Ra ratio and actual Ra ratio. The absolute value of relative errors in five groups is from 5.64% to 13.47%. The relative errors are within the allowable range that proves the accuracy of the roughness prediction model.

**Table 7.** Results of polishing experiments for model verification.

| No. | Polishing Pressure (N) | Rotation Speed of Tool (r/min) | Feed Speed (mm/s) | Predicted Ra Ratio | Actual Ra Ratio | Relative Error |
|---|---|---|---|---|---|---|
| 1 | 46 | 622 | 1.01 | 0.454 | 0.481 | −5.64% |
| 2 | 30 | 1860 | 1.19 | 0.485 | 0.528 | 8.88% |
| 3 | 41 | 804 | 0.76 | 0.382 | 0.434 | −12.03% |
| 4 | 43 | 1447 | 1.25 | 0.472 | 0.416 | 13.47% |
| 5 | 50 | 1593 | 0.25 | 0.248 | 0.238 | −9.66% |

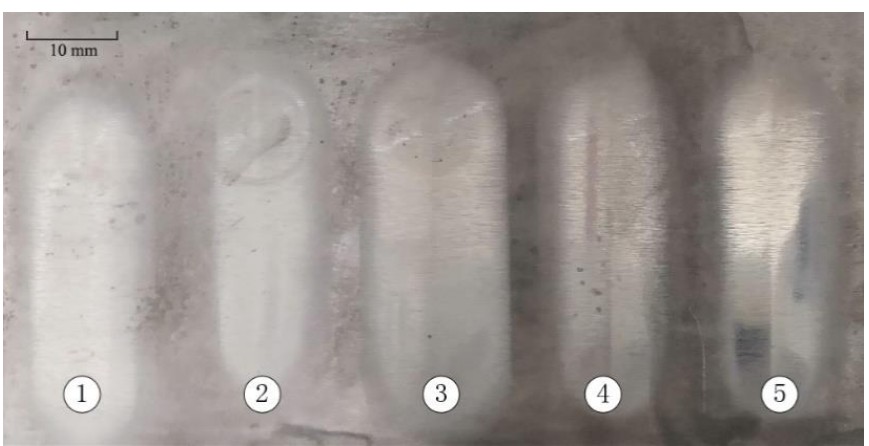

**Figure 10.** The mold steel surface of polishing experiments verification.

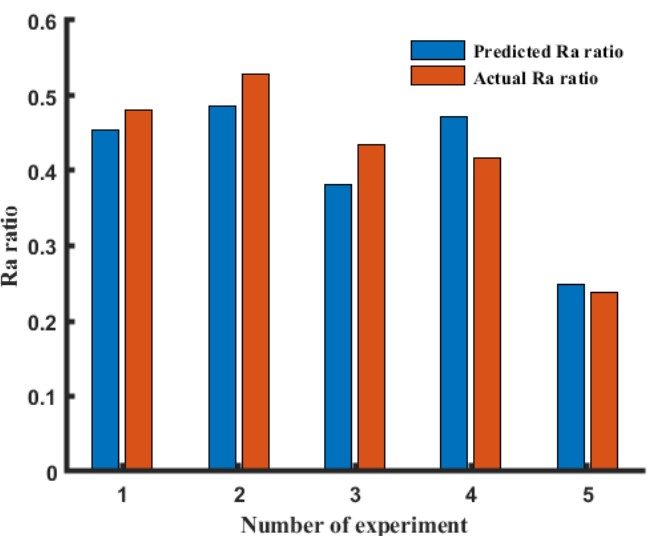

**Figure 11.** The comparison of predicted Ra ratio and actual Ra ratio.

## 4. Experiment Validation of Multipath Polishing

In the previous section, the parameters optimization of a single polishing path has been carried out through establishing the model of roughness prediction. In practical machining, it is necessary to plan the polishing path and combine the optimal parameters of a single path to finish whole surface polishing of mold steel.

The polishing path is planned according to the shape of the workpiece surface, and polishing paths commonly used by robots mainly include raster path, spiral path, etc. [21,22]. The object of polishing is plane mold steel, so raster path is selected to plan polishing path. The spacing of raster path is an important factor affecting polishing accuracy and efficiency. In order to verify the effectiveness of optimal polishing parameters for whole surface polishing, raster paths with different spacing are applied to experiments of whole surface polishing. Five pieces of the same area ($90 \times 50$ mm$^2$) from the same mold steel workpiece were selected, and five groups of raster paths with different spacing according to the radius of the polishing tool ($r = 16.5$ mm) were set to conduct polishing experiments. Figure 12 shows the surface of the mold steel with different polishing paths. The optimal process parameters (polishing pressure of 50 N, rotation speed of tool of 1600 r/min, feed speed of 0.25 mm/s) were adopted in polishing experiments. The experimental results with different polishing paths are demonstrated in Table 8.

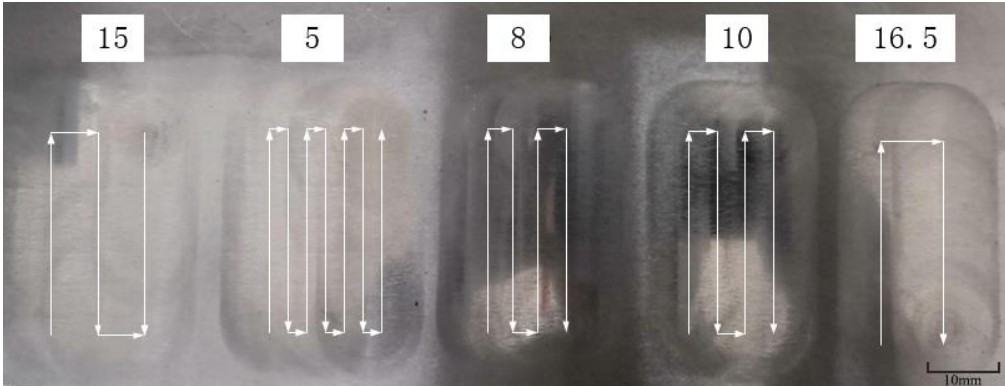

**Figure 12.** The surface of mold steel with different polishing paths.

**Table 8.** The experimental results with different polishing paths.

| Spacing/mm | 16.5 | 15 | 10 | 8 | 5 |
|---|---|---|---|---|---|
| Number of paths | 2 | 3 | 4 | 4 | 7 |
| Time/min | 9.1 | 14.0 | 17.8 | 17.6 | 29.8 |
| Initial Ra/μm | 0.941 | 0.885 | 0.914 | 0.986 | 0.944 |
| Ra after polishing/μm | 0.334 | 0.315 | 0.275 | 0.307 | 0.290 |
| Ra ratio | 0.355 | 0.356 | 0.301 | 0.311 | 0.307 |

The surface roughness decreases to about 0.3 μm effectively after polishing with the optimal process parameters (the expected roughness ranges from 0.2 to 0.4 μm.), which proves the feasibility of optimization for process parameters. The experiment group with a better effect is the spacing of 10 mm, the surface roughness of 0.275 μm and Ra ratio of 0.301. The variation trend of surface roughness and Ra ratio corresponding to different polishing spacing is exhibited in Figure 13. It is easy to see that with the decrease in the polishing spacing the surface roughness firstly decreases and then increases, and the overall trend decreases. However, too small spacing will increase the polishing time and lower the polishing efficiency. Additionally, it is necessary to select a suitable polishing path for polishing by considering the actual requirements of machining accuracy and efficiency.

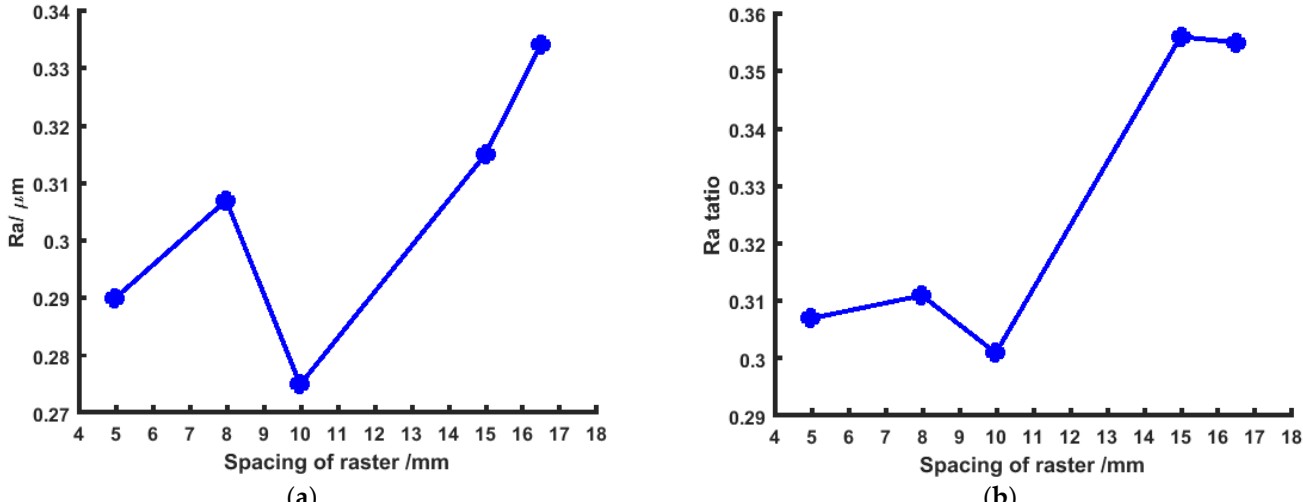

**Figure 13.** The variation trend of surface roughness and Ra ratio with different spacing. (**a**) Surface roughness; (**b**) Ra ratio.

## 5. Conclusions

1. A robotic polishing platform with force control was built to realize the automatic polishing of mold steel. The influence rule of robotic polishing parameters, which included polishing pressure, rotation speed of tool and feed speed on surface roughness was analyzed through a single-factor experiment, and the parameters' range of central composite experiment was determined.

2. The prediction model of surface roughness was established through the center composite design experiment. The prediction model that improved by variance analysis is significant. Additionally, the polishing experiments were conducted with five groups of process parameters and combinations were randomly selected. The relative errors of predicted Ra ratio and actual Ra ratio are within the allowable range (the maximum error value is 13.47%). that proves the accuracy of roughness prediction model.

3. The optimum polishing parameters were achieved according to the response surface method. Additionally, experiments of multipath polishing were conducted to verify the feasibility of optimization for polishing parameters. The surface roughness decreased to about 0.3 µm effectively after polishing with the optimal process parameters. The prediction model of surface roughness and optimum polishing parameters are helpful to improve surface quality in robotic polishing for mold steel.

**Author Contributions:** Methodology, Y.X. and J.L.; simulation analysis, Y.X. and G.C.; data curation, J.Y.; validation, Y.X., J.Y. and G.C.; writing-original draft preparation, Y.X. and G.C.; writing-review and editing, J.L.; experiment operation, G.C. and M.Z.; funding acquisition, J.L. All authors have read and agreed to the published version of the manuscript.

**Funding:** This research was funded by Scientific and Technological Project of Quanzhou (No.2020C071, No. 2021C021R and 2019STS011).

**Institutional Review Board Statement:** Not applicable.

**Informed Consent Statement:** Not applicable.

**Data Availability Statement:** Not applicable.

**Acknowledgments:** This research was supported by Laboratory of Robotics and Intelligent Systems (CAS Quanzhou) and Beijing Key Laboratory of Advanced Manufacturing Technology.

**Conflicts of Interest:** The authors declare no conflict of interest.

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
