# Peer review of "Process Optimization of Robotic Polishing for Mold Steel Based on Response Surface Method"

_machines, doi:10.3390/machines10040283_

Round 1

Reviewer 1 Report

Very interesting topic. In this paper the focus is put on the study that optimises several process parameters for robotic polishing, including polishing pressure, feed speed, and tool rotation speed. If possible, figures 7 - 9 could be in higher resolution, please increase the resolution for mentioned pictures. I wish you much success in your further research.

Reviewer 2 Report

The paper entitled "Process optimization of robotic polishing for mould steel based 2 on response surface method" describes the surface quality after polishing of 40Cr steel using a robot.

The results presented are interesting and address an important topic in the field of finishing machining. The paper is coherent, and has a clear layout. The abstract sufficiently captures the description of the research work.

The introduction provides a correct description of the topic. The subject matter, scope, and choice of scientific publications cited are adequate to the topic discussed.

However, there are gaps in the description of the research methodology and editorial errors that should be corrected.

There is no paragraph clearly emphasizing the novelty aspect of the presented work. In recent years, there has been a lot of publications on robotic polishing. The aspect of novelty needs to be specified. What makes the proposed research stand out and what does it add to this field of science.

Line 55: This work is not about polishing. Citation of this work is not related to the topic of the article.

Line 96: Specify the other parameters of the robot, in particular the accuracy.

Line 109: State the brand and model of the servomotor.

Line 120: What condition was the steel in? Was it hardened?

Line 124: What was the speed of the measurement? What was the diameter of the tip? How long was the cut-off?

Line 148: Were the points measured centrally in the middle of the path? In which direction was the measurement carried out?

Line 155: The article mentions pressure and the values are given in Newtons.

Figure 5: What was the initial roughness of the samples?

Table 3: Why was the whole range of parameters from the previous experiment used? Should not some of the values be excluded?

Line 281: What does ‘radio’ mean here? Did you mean ‘ratio’?

Figure 12: Where were the measurements taken?

There is no scale on Figures 3, 6, 12.

The article is of a high scientific standard with good statistical analysis. Once the authors make some improvements, I can recommend the article for publication.

Reviewer 3 Report

Authors presented research on improving surface quality of mold steel by reducing surface roughness (Ra) on the way of the optimizations of process parameters for robotic polishing. 
The influence of the following control parameters: polishing pressure, feed speed and rotating speed of tool was tested in the research. 
The results of experiments show that the surface roughness decreased significantly after polishing with the optimum polishing parameters. English and US English are used in the paper. This aspect should be harmonized.

The state of art analysis is too modest, and no conclusions can be drawn showing the reasons for conducting the research. In this state, the question can be asked: Why was this analysis done? It should be extended, for example, with papers presenting UAN in other erosive processes, such as:
https://doi.org/10.1063/1.5092054
http://casopisi.junis.ni.ac.rs/index.php/FUMechEng/article/view/7427/4109
and others.

The information contained in the last paragraph of Chapter 1. Introduction is more suitable for the Abstract.

noticed errors:

The names of the authors are written with the first capital letter
The spelling of the word mold should be standardized.
Table 1, last column: What is: Kg /m^3?
We put a space between the value and the unit, for example:
line 110 - it is 400W, it should be 400 W
We express the roughness value in micrometers. The abbreviation is µm, not um.

Reviewer 4 Report

The manuscript "Process optimization of robotic polishing for mould steel based on response surface method" provides an analysis of surface roughness and surface improvement for selected three parameters of the polishing process. The proposed analytical model for the prediction of surface roughness was validated using variance analysis. An accurate fit between predicted and experimental data was obtained for the assumed conditions. The scope of the studies and analyses carried out is basic. The applicability of the included data is limited due to the study of one material. What surface roughness is recommended for the selected steel application? It would be necessary to compare the data values obtained as a result of the proposed optimisation with the expected value. For a better explanation of the phenomena, please consider the following comments:

Page 3, Section 2.2: The tested material should be described in detail (chemical composition, average roughness).

Page 5, Figure 5: The figures relate to the analysis of two variables assuming the effect of three. What values were assumed for the third variable?

Page 6, Figure 6: The photos appear to have been taken under different lighting conditions. The differences in the polished surface are difficult to determine. It is advisable to prove the differences for higher magnification.

Page 13, Section 5: The conclusions should include numerical data to prove the statements.

Round 2

Reviewer 3 Report

Most of my little suggestions have been corrected. However, the analysis of the literature has been extended to only one not very adequate item. I suggested that the literature review should be enriched with at least 2 other items containing the RSM methodology, while the authors stubbornly have added only one and not necessarily on the topic.
I consider this to be a disregard for my comments and I am against publishing the paper in this form. In addition, I would like to note that, there are also other countries where scientists use Response Surface Methodology (RSM) and publish results in this regard. It seems to me that the poorly understood local patriotism shouldn't have in objective, impartial scientific works.

Author Response

Dear reviewer,

I am sorry that the previous answer was not appropriate. We have added four articles covering the RSM approach, including the two items you suggested earlier. We thank you very much for your enthusiastic work, and your comments will help us to further improve. If there are any other changes we can make, we would very much like to make them and we would appreciate your help.

Sincerely yours,

Xie Yinhui

Reviewer 4 Report

Thank you for your responses. The revised manuscript considers the reviewer's comments.

Author Response

We are very grateful for your warm work earnestly. In all, we found your comments are quite helpful. They point the deficiencies about our manuscript us, and help us for the further improvement.